# Individual quality, insecure organizational attachment, and formalistic task completion: Social cognitive perspective

Wenjun Wu[1], Huan Xiao[2], Dengke Yu[1]*

**1** School of Public Policy and Administration, Nanchang University, Nanchang, China, **2** Postdoctoral Research Station of Management Science and Engineering, Nanchang University, Nanchang, China

* yudengke@ncu.edu.cn

**Data Availability Statement:** All relevant data are within the manuscript and its Supporting information files.

## Abstract

Formalistic tasks are widely utilized in modern companies due to their ability to increase productivity and contribute to the achievement of corporate goals at a lower cost. However, these tasks are often meet with resistance from individuals because they do not provide direct short-term rewards for their efforts. Drawing on social cognitive theory, this study examined the influence of individual quality and organizational attachment on the completion of formalistic tasks. To address this, the study conducted a questionnaire survey to collect data from 602 Chinese respondents and built a structural equation model for data analysis. Through empirical research, the study confirmed the positive role of individual quality, including knowledge and personality, in the completion of formalistic tasks. Furthermore, the study proved that avoidant attachment could significantly weaken the effect of some components of individual quality on formalistic task completion. This paper is the first to reveal the influence of individual and environmental factors on individuals' completion of formalistic tasks, progressing from bottom to top. The implications of these results are discussed.

## Introduction

The formalistic tasks investigated in this study encompass normative, mandatory, and bureaucratic tasks that are hierarchically mandated, executed by subordinates, and aligned with the formal directives prescribed by their superiors. These tasks, as perceived by subordinates, typically possess the following characteristics. Firstly, they are assigned by superiors rather than being undertaken voluntarily. Finally, their superiors are more likely to require to "sound good" than "do good" when they come to formalistic task completion [1]. Secondly, these tasks typically bear no relevance to their job requirements, and the successful completion thereof holds no sway over their recent work performance. Thirdly, individuals engaging in these tasks often exhibit a lack of awareness or concern regarding their organizational significance; however, they do recognize that such tasks do not directly contribute to their career development or personal progress. Lastly, it is noteworthy that superiors are more inclined to prioritize the appearance of competence ("sound good") over substantive effectiveness ("do good")

**Funding:** This work was funded by National Natural Science Foundation of China, grant number 71962021 and 72362028.

**Competing interests:** The authors declare that they have no conflicts of interest to report regarding the present study.

when overseeing the formalistic completion of tasks [1]. Driven by negative cognitions, formalistic tasks often pose challenges in achieving high effectiveness and efficiency. The resistance and excessive energy consumption not only diminishes their satisfaction with the organization but also undermines operational efficiency [2]. This detrimental impact is associated not only with the nature of the formalistic tasks themselves but also with the manner in which they are structured. Specifically, the hierarchical power disparity between superiors and subordinates, who are compelled by individual superiors to undertake formalistic tasks, intensifies pressure on the subordinates. This heightened pressure can potentially provoke counterproductive behaviors among subordinates, particularly in contexts characterized by elevated formalism.

Despite the problems inherent in formalistic tasks, they are deemed essential, rational, and valuable within collectivist cultures [3]. However, their value tends to be more collective than individual, more indirect than direct, and more spiritual than material. The execution of formalistic tasks contributes to the cultivation of organizational culture, the establishment of organizational order, and the pursuit of organizational interests [4]. Through participation in formalistic tasks, subordinates gradually develop a sense of support for superior directives and identification with the organization. This fosters a consensus among individuals that their personal interests should align with collective interests [5].

While it might be perceived as encroaching on individual rights and freedom within an individualistic organization, the collectivist perspective views the individual not as an isolated, rational, and competitive entity separate from the group, but as an integral part of the collective whole [6]. Upholding the collective and realizing collective interests indirectly safeguards and enhances individuals' rights and interests in the long run.

Formalistic tasks exhibit pros and cons. This paper endeavors to affirm their value rather than magnify their flaws, marking a departure from previous research approaches. The distinctive element of this study lies in proposing a new concept. The management of formalistic tasks differs from similar proposals found in prior literature, such as illegitimate tasks [7], symbolic management [8], performative governance [1], and bureaucratic management [9]. Past research on these concepts has predominantly focused on their negative implications, seldom delving into their positive value. This study suggests that mitigating the negative effects of formalistic tasks is achievable through some measures. Improvement lies not only in optimizing the form and content of tasks issued by superiors but also in enhancing subordinates' cognitive and support for these tasks. This enhancement, in turn, relies on improving subordinates' quality and reducing their insecure attachment to the organization. Consequently, we propose that exploring the mechanism through which subordinates' individual quality and organizational attachment influence formalistic task completion is crucial. This exploration guides us towards enhancing the approval rate and completion effectiveness of formalistic tasks by improving the interaction between individuals and their organizational environments.

The study focused on addressing two key research questions. Firstly, it examined how the components of individual quality impact the completion of formalistic tasks. Secondly, it delved into the role of organizational attachment in the mechanism of completing formalistic tasks. The Theories section introduces the basic theories, while the Literature Review section provides a review of prior literature. In the Hypotheses section, the development of hypotheses was presented. The methodology, encompassing sample selection, data collection, measurement, and statistical techniques, was detailed in the Methodology section. The Results section displayed the results, followed by a discussion of implications, limitations, and future research in the Discussion section. Finally, the Conclusion Remark section summarized the conclusions.

## Theories

Social cognitive theory posits that an individual's behavior is driven by their cognition, shaped through the reciprocal interaction between the individual and their living environment [10]. The individual functions as both the sculptor and the product of this environment. When individuals perceive increasing incentives from their surroundings, they show a willingness to exert more effort and provide more positive, high-quality feedback towards a particular behavior [11]. In the realm of formalistic task completion, subordinates, upon recognizing the task's value to both the organization and themselves, demonstrate support and cooperation in completing the task [12]. Contrarily, those who resist will opt to complete tasks at the lowest cost while precisely meeting their superiors' requirements. Additionally, their support levels and completion rates for formalistic tasks will fluctuate in tandem with changes in their insecure organizational attachment [13]. Building on this, Malik et al. [14] proposed that prolonged fear and panic can diminish employees' passion for their work, eroding individual motivation to pursue excellence and efficiency. This can lead to heightened job insecurity among individuals, further compromising performance as it weakens their commitment to organizational goals. Consequently, we have distilled the critical influencing factors of formalistic task completion into two clusters: individual quality from an individual perspective and insecure organizational attachment from an environmental perception perspective.

The variations in individual quality contribute to diverse cognitive styles, abilities, perceptions of the environment, and societal learning, ultimately resulting in distinct behaviors [15]. According to the three-layered onion model developed by Curry [16], individual differences in quality manifest across three levels, akin to the layers of an onion: an inner 'cognitive personality' layer, a middle 'information processing' layer, and an outer 'instructional preference' layer. The inner layer encapsulates deep and enduring personalities, influencing how individuals assimilate and adapt information to comprehend reality. The middle and outer layers include information, knowledge, and skills [16]. This onion model, underscoring the significance of personality and knowledge in representing individual differences, has evolved into a widely cited integrated model in the field of cognitive styles [17, 18]. Numerous studies have stated and substantiated the impact of individuals' knowledge and personalities on the completion and performance of work tasks [19, 20]. As a result, this study posited that the combined influence of individuals' personalities and knowledge would shape their perception of the value, cost, and risks associated with formalistic tasks, subsequently influencing their decisions, behaviors, and task performance.

Attachment theory, as a component of cognitive theory, focuses on the cognitive-affective processes related to "attachment." This term is defined as the human inclination to seek and foster emotional connections with specific individuals [13]. Currently, this theory offers a unique relational perspective that contributes to the exploration of organizational behavior. Organizational attachment is categorized into secure and insecure types. High levels of secure attachment, along with low levels of insecure attachment, yield positive organizational outcomes, such as proactive work behavior and high-quality leader—member exchange [21, 22]. Scholars have directed their attention towards understanding and managing insecure attachment, characterized by negative emotions such as anxiety or distrust towards the organization. Insecure attachment is further divided into two dimensions: anxious attachment and avoidant attachment [23]. The former pertains to the extent to which an individual worries about the availability of others in times of need, seeking their love and care anxiously. The latter refers to the extent to which an individual distrusts the goodwill of others, defensively striving to maintain behavioral and emotional independence [24]. Formalistic tasks are frequently completed within a group context. The evaluation of the value and risk associated with these tasks is

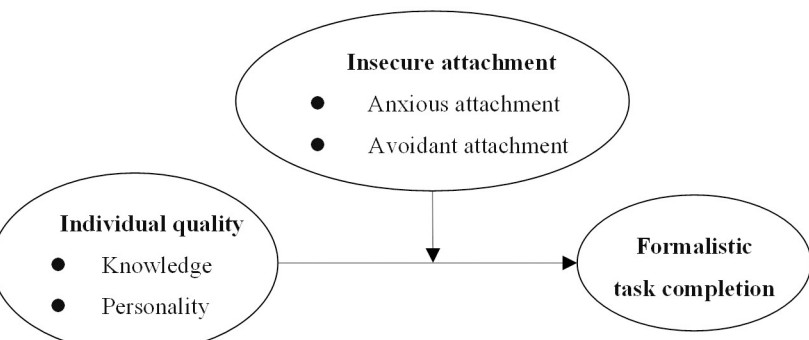

**Fig 1. The framework of this study.**

inevitably influenced by both the organization and its members. When an individual shows a higher degree of insecure attachment to others, their cognition and decision-making are more likely to be intervened by negative emotions in the context. Consequently, insecure attachment weakens an individual's engagement and completion of tasks.

We then constructed the study framework as follows, shown in Fig 1.

## Literature review

### Research on individual quality

This paper defines individual quality as a collection of styles and abilities embedded in an individual's body, distinguishing one person from another and influencing positive outcomes. It is distinct from individual cognitive style, which relies on both individual reserve and the processes of information acquisition, processing, and exploitation [25]. Nonetheless, the study employs the onion model, a theory of cognitive style, to analyze the components of individual quality. This construct focuses on the individual's reserve, directing the study to the field of one's knowledge and personality. The focus is concerned by psychologists who have explored the intertwined dynamics of knowledge and personality in the research on human creativity and behavior [19, 26].

Knowledge plays a pivotal role in shaping individual behavior through three mechanisms. First, it functions as a determinant in the decision-making process. Strang et al. [27] stated that an individual's cognitive ability and knowledge about a task significantly impacted group decision-making performance. Second, knowledge acts as an input for generating innovative ideas. According to Santangelo and Phene [28], the diversity in individual knowledge is crucial for enhancing organizational creativity performance. Enterprises could bolster their performance generation capabilities by encouraging the sourcing and sharing of knowledge. Third, knowledge serves as a process of knowledge transfer. Many scholars have examined the positive effects of the entire knowledge transfer process, including aspects such as knowledge learning [29], sharing [30], and creation [31], on individual or organizational outcomes.

Personality refers to the totality of an individual's relatively stable psychological characteristics, behavioral patterns, and emotional responses [32], spanning features in cognition, emotion, and behavior. It is distinguished by stability, consistency, and predictability [33, 34]. One of the most well-known personality theories is the Big Five model, which categorizes personality into five dimensions: extroversion, openness, conscientiousness, agreeableness, and neuroticism [35]. In previous studies, extroversion and openness personalities had demonstrated

positive associations with individual motivation and behaviors, as they fostered the establishment of social relationships and the development of social capital, helping individuals get support from society [36]. The effects of conscientiousness and agreeableness are two-fold. On one hand, they contribute positively to the performance of tasks in stable environments [37]. On the other hand, these traits may impede individuals and organizations from engaging in creative activities and negatively affect their responses to dynamic and evolving environments [38, 39]. Considering neuroticism as a negative personality trait, scholars often examine the positive effects of its reverse counterpart, namely emotional stability. For instance, Eichel and Stahl [40] proposed that the positive relationship between emotional stability and mindfulness enhance individuals' performance in error detection tasks.

## Research on insecure organizational attachment

Organizational attachment is classified into secure attachment and insecure attachment [41]. Individuals with secure attachment feature positive self-evaluations and positive evaluations of others. In contrast, insecure attachment involves negative evaluations and stressful relationships with others [42]. Insecure attachment further includes anxious attachment (negative self-evaluation and positive other-evaluation), avoidant attachment (positive self-evaluation and negative other-evaluation), and a mixed type combining negative self-evaluation and negative other-evaluation [43]. In the organizational context, employees with secure organizational attachment show high levels of self-confidence and trust in the organization, their leaders, and colleagues [22]. while employees with insecure organizational attachment manifest anxiety concerning working relationships and performance, and encounter conflicts with colleagues [43–45].

Individuals with high levels of anxious organizational attachment tend to remain silent when their opinions differ from those of the group, superiors, or even peers. Instead, they choose to conform to the decisions of the majority [46]. Conversely, those with high levels of avoidant organizational attachment, who predominantly trust their own judgments, show a lack of team spirit. They engage in minimal communication with others and are unwilling to prioritize collective interests over individual ones [47]. Hence, both anxious and avoidant attachments lead to interpersonal deviance behaviors, ultimately harming task performance. Vîrgă et al. [44] proposed that individuals with insecure attachments tend to be sensitive to interpersonal relationships, potentially engaging in interpersonal transgressions. Empirical evidence supported the notion that insecure attachment was positively associated with job burnout and negatively associated with job performance. Kirrane et al. [48] stated that insecure attachment significantly impairs employees' perceptions of the quality of "leader-member exchange" and "team-member exchange" relationships, resulting in diminished creative outputs. Previous research has consistently highlighted the role of insecure organizational attachment as a moderating factor, indirectly predicting workplace deviance behaviors [43, 49].

## Research on formalistic task and its completion

Following an extensive literature review, scant scholarly works were identified that offer a clear definition of formalistic tasks (or work) and formalism (or formalism). It is a local phenomenon that observed in the context of eastern collectivist culture and system [3, 4]. Formalistic tasks can be characterized by a combination of features, including extra-role elements, unexpected demands, momentary requirements, mandatory aspects, bureaucratic procedures, and tasks considered illegitimate [50–52]. From a comprehensive perspective, these tasks can be understood in three dimensions. First, in terms of form, they are contingently generated from top to bottom, catching subordinates off guard. Second, in terms of content, they are loosely

associated with the duties of subordinates, thereby establishing a vague and indirect causal relationship with their performance. Lastly, concerning completion, a significant portion of subordinates tends to mentally resist and provides minimal support in participating and completing these tasks. Due to these characteristics, the majority of employees show reluctance to take the initiative in completing formalistic tasks most of the time [53].

While most of studies have focused on task performance [e.g., 54–56], this study suggests a shift in focus towards the completion of formalistic tasks. The rationale lies in the often long-term, indirect, and collective impact of such tasks, making their assessment generally difficult. The key to improving the completion of formalistic tasks lies in task transformation, and two approaches can be employed: the procedural way and the affective way. The procedural way entails refining the assignment process to render formalistic tasks more legally sound, reasonable, valuable, and intriguingly, even leisure-oriented. This transformation aims to ensure that such tasks are embraced by the majority of subordinates [57]. The affective way recommends enhancing subordinates' perception of formalistic tasks through improved supervisor-subordinate communication and emotional support from coworkers. This approach is intended to foster a better understanding of the value and necessity of these tasks, encouraging subordinates to complete them voluntarily [57].

## Hypotheses

### The relationship between individual quality and formalistic task completion

Individual knowledge has positive effect on the completion of formalistic tasks through three mechanisms. Firstly, the knowledge about the task shapes individuals' perception and evaluation of its value, cost, and associated risks during participation and completion. Greater information and knowledge empower individuals to make more accurate decisions, directly influencing their behavior [58]. Secondly, the level and structure of individual knowledge play a pivotal role in shaping their self-confidence and self-efficacy throughout the task completion process. A diverse range of task-related knowledge facilitates better organization, promotion, and successful completion of formalistic tasks by teams [59]. Lastly, knowledge-related behaviors, such as learning and sharing, contribute to improving connections and trust among individuals. This enhancement proves beneficial for creating emotional incentives within the group, thereby fostering collaborative efforts toward the completion of formalistic tasks [60].

Individual personalities make a positive effect on the completion of formalistic tasks. Firstly, as indicated by the research of Yu et al. [61] and Zhang et al. [62], employees with high levels of proactive personality are more inclined to recognize the collective value and underestimate their ego depletion when completing formalistic tasks. This tendency contributes to the emergence of organizational citizenship behavior and an enhancement in task performance. Similarly, Fila and Eatough [63] proposed that engagement in illegitimate tasks can increase stress, anxiety, and emotional exhaustion, with individuals' perception of these stressors linked to their psychological characteristics. Consequently, individuals with positive personalities are more likely to accept and complete formalistic tasks. Secondly, the specific personality compositions within teams, such as conscientiousness and emotional stability, had been shown to foster individual work motivation and emotional regulation, respectively, aligning with positive performance [64]. Finally, individuals possessing higher levels of the Big Five personalities exhibit greater inclusivity and adaptability, leading to heightened performance in citizenship behaviors and general tasks [36]. This adaptability renders them more confident, altruistic, and proficient in transferring their skills from a general task to another momentary task [51]. Consequently, their satisfaction with formalistic tasks sees an increase.

We therefore proposed the following hypothesis. **H1**: Individual quality (a. knowledge and b. personalities) has a positive effect on formalistic task completion.

## The moderating role of insecure organizational attachment

A limited number of individuals display active willingness to complete formalistic tasks due to the aforementioned characteristics. A task that is actively accepted and completed deviates from the definition of a formalistic task. Therefore, external incentives, such as social support and leadership encouragement, play a crucial role in motivating subordinates to complete formalistic tasks [63]. Specifically, leaders and colleagues within organizations exert influence on the decisions and behaviors of individuals engaged in formalistic tasks. The extent of this influence is contingent upon an individual's level of insecure organizational attachment.

Individuals with high levels of avoidant attachment tend to place trust in themselves while harboring a distrust towards others. Hence, they are inclined to question the long-term and indirect benefits of formalistic tasks assigned by the organization, especially when lacking a clear perception of these advantages. In such instances, they remain impervious to persuasion from the organization and their superiors, opting instead for avoidance or engaging in self-protective behavior [65]. On the contrary, individuals characterized by high anxious attachment tend to trust others but lack confidence in themselves. As a result, they often emulate the behaviors of their colleagues and conform to the decisions made by their superiors, revealing a mechanism of facades of conformity [66]. However, in the specific context of completing formalistic tasks, the behaviors of colleagues and decisions of superiors may potentially diverge. Consequently, the impact of anxious attachment on individuals' inclination to either support or resist formalistic tasks remains uncertain, contingent upon the prevailing decision within the organizational majority.

The effects of individual quality and insecure organizational attachment on the completion of formalistic tasks is interactive. Individual quality serves as an internal impetus, and insecure organizational attachment acts as a stimulus from an environmental perspective. Therefore, it can propose that insecure organizational attachment functions as a moderator. Whether characterized by avoidant or anxious attachment, both reveal an abnormal relationship between the individual and the organization, as well as with other members. This abnormality results in heightened mental and energy consumption for the individual, leading to an increased psychological burden [47]. As a consequence, this dynamic weakens the otherwise positive effect of individual quality.

The following hypothesis was proposed. **H2**: Insecure organizational attachments (a. avoidant attachment and b. anxious attachment) negatively moderate the relationship between individual quality and formalistic task completion. Among the moderating mechanism, the direct effect of avoidant attachment is significantly negative, but the direct effect of anxious attachment is not significant.

## Methodology

### Data

The data was collected for student respondents gathered through the following three channels. Firstly, the questionnaire was launched on the Sojump website (http://www.sojump.com), a professional Chinese institution specializing in online questionnaire surveys. The targeted respondents comprised college students registered nationwide as members of the institution, and 219 respondents were randomly selected from its database between June 27th and July 4th, 2022. A monetary incentive of 7 yuan (approximately 0.75 dollars) was provided to the participants. Secondly, the survey was published on various media outlets in China from May

**Table 1. Means, standard deviations and correlations.**

| Variables | Mean | SD | Correlations | | | | | | | | |
|---|---|---|---|---|---|---|---|---|---|---|---|
| | | | 1 | 2 | 3 | 4 | 5 | 6 | 7 | 8 | 9 |
| 1 Knowledge | 3.296 | 0.729 | 0.727 | | | | | | | | |
| 2 Extroversion | 3.487 | 0.797 | 0.276** | 0.722 | | | | | | | |
| 3 Openness | 3.645 | 0.665 | 0.334** | 0.534** | 0.688 | | | | | | |
| 4 Conscientiousness | 3.930 | 0.698 | 0.264** | 0.393** | 0.460** | 0.697 | | | | | |
| 5 Agreeableness | 3.930 | 0.613 | 0.211** | 0.498** | 0.525** | 0.519** | 0.692 | | | | |
| 6 Emotional stability | 3.359 | 0.737 | 0.254** | 0.510** | 0.517** | 0.404** | 0.382** | 0.684 | | | |
| 7 Avoidant attachment | 2.997 | 0.750 | -0.129** | -0.120** | 0.021 | -0.073 | -0.021 | -0.030 | 0.656 | | |
| 8 Anxious attachment | 3.182 | 0.837 | -0.144** | -0.087 | 0.040 | 0.004 | 0.146** | -0.080 | 0.482** | 0.754 | |
| 9 Formalistic task completion | 3.786 | 0.657 | 0.252** | 0.339** | 0.364** | 0.424** | 0.462** | 0.251** | -0.133** | -0.013 | 0.773 |

n = 465,

***$p<0.001$,

**$p<0.010$,

*$p<0.050$.

The diagonal values are the square root of AVEs.

18th to July 10th, 2022. Invited respondents voluntarily and freely responded to the questionnaires, and these individuals included the schoolmates and friends of the authors, as well as their schoolmates, aligning with the principles of snowball sampling. Thirdly, the questionnaire was distributed offline to college students at Nanchang University on May 17th and 18th, 2022. Researchers encountered these students in locations such as study rooms or the library. In total, 602 respondents were invited; however, 162 questionnaires were incomplete or invalid. Therefore, 465 valid questionnaires were collected, resulting in a 77.24% effectiveness rate. Following Hair et al.'s [67] recommendation that a minimum of 300 samples is necessary for estimating a structural equation model with seven constructs or fewer, the sample size in this study satisfied the stipulated requirement.

The demographic characteristics of our sample were as follows. Males comprised 38.7%, while females constituted 61.3% of the respondents. Regarding age distribution, 95.1% of participants fell in the 15 to 26 age range, 4.8% were aged between 27 and 31, and only 0.2% belonged to other age groups.

Regarding educational level, 76.10% were undergraduates, 13.10% were master's students, 1.30% were doctoral students, and the remaining 9.5% were students at other academic levels. In terms of political status, 17.80% of respondents were members of the Communist Party of China (CPC), 32.30% identified as part of the masses, and the remaining 49.9% held other political status.

The means and standard deviations (SD) of the constructs, along with the correlations among them, were calculated and are presented in Table 1. The mean values indicate that individual personalities and formalistic task completion are relatively high, while individual knowledge and organizational attachments are comparatively low. The deviations, when compares to the means, are not substantial. The data basically complied with the requirements of homogeneity and normality. Additionally, both knowledge and personalities show positive correlations with formalistic task completion, whereas insecure organizational attachments display a negative correlation with formalistic task completion. Notably, the correlation between anxious attachment and task completion is not statistically significant. The identified correlations substantiate the relationships among the core variables in this study and provide

support for subsequent hypothesis testing. Furthermore, the square roots of Average Variance Extracted (AVEs) surpass the correlations between the constructs, indicating that the study's data meet the discriminant validity criterion as suggested by Fornell and Larcker [68].

## Measures

The questionnaire involved the measurement of five variables: knowledge, personalities, avoidant attachment, anxious attachment, and formalistic task completion. Among these, individual knowledge and personalities served as independent variables, avoidant and anxious attachments functioned as moderating variables, and formalistic task completion represented the dependent variable. All variables were measured by the 5-point Likert scale, ranging from 1 (strongly disagree) to 5 (strongly agree).

Individual knowledge was measured by 4 items that were developed by Yu and Wu [69]. A sample item was, "In the past six months, I had frequently read various books." The scale yielded a Cronbach's alpha of 0.816. Evaluating individual personalities involved a simplified "Big Five" personality test, incorporating 15 items developed by Meng et al. [70]. These items measured Chinese individuals' extroversion, openness, conscientiousness, agreeableness, and emotional stability. A sample item measuring conscientiousness was, "In work and study, I will complete tasks promptly." This personality scale yielded a Cronbach's alpha of 0.882. Anxious and avoidant attachments were measured by four and three items, respectively, which were developed by Feeney et al. [23]. Sample items included, "I worry that my organization doesn't care for me," and "I find it difficult to allow myself to depend on my organization." These two scales yielded Cronbach's alpha of 0.837 and 0.693 respectively. The measurement of formalistic task completion occurred in a pretest university context, commonly regarded as a formalistic task by most students. In this scenario, respondents were compelled to engage in a tedious and worthless lecture activity unrelated to their professional development. The items were adapted from the scale developed by Van Dyne and LePine [71], focusing on the construct of extra-role task performance. A sample item was, " I can complete it at the level expected by my superiors" The scale yielded a Cronbach's alpha of 0.854.

**Reliability, validity, and common method bias.** The study conducted a confirmatory factor analysis to assess the discriminant and convergent validity of the scale, as shown in Table 2. The guidelines outlined by Podsakoff et al. [72] were extended for assessing model fit. A single-factor model was applied, resulting in a poor fit for the data. And the hypothesized

**Table 2. The result of confirmatory factor analysis.**

| Models | $\chi^2$ | df | $\chi^2$/df | RMSEA | CFI | TLI | SRMR |
|---|---|---|---|---|---|---|---|
| **Hypothetical model** with nine factors | 557.004 | 369 | 1.509 | 0.033 | 0.964 | 0.957 | 0.040 |
| **Eight factor model** (Knowledge, Extroversion, Openness, Conscientiousness, Agreeableness, Emotional stability, Avoidant attachment + Anxious attachment, Formalistic task completion) | 686.549 | 377 | 1.821 | 0.042 | 0.941 | 0.931 | 0.046 |
| **Seven factor model** (Knowledge, Extroversion + Openness, Conscientiousness, Agreeableness, Emotional stability, Avoidant attachment + Anxious attachment, Formalistic task completion) | 787.465 | 384 | 2.051 | 0.048 | 0.923 | 0.912 | 0.050 |
| **Six factor model** (Knowledge, Extroversion + Openness, Conscientiousness + Agreeableness, Emotional stability, Avoidant attachment + Anxious attachment, Formalistic task completion) | 897.799 | 390 | 2.302 | 0.053 | 0.903 | 0.891 | 0.053 |
| **Five factor model** (Knowledge, Extroversion + Openness + Conscientiousness + Agreeableness, Emotional stability, Avoidant attachment + Anxious attachment, Formalistic task completion) | 1046.707 | 395 | 2.650 | 0.060 | 0.875 | 0.862 | 0.058 |
| **Four factor model** (Knowledge, Personality, Avoidant attachment + Anxious attachment, Formalistic task completion) | 1154.942 | 399 | 2.895 | 0.064 | 0.855 | 0.842 | 0.062 |
| **Three factor model** (Individual quality, Insecure organizational attachments, Formalistic task completion) | 1643.462 | 402 | 4.088 | 0.081 | 0.762 | 0.742 | 0.078 |
| **Two factor model** (Individual quality + Insecure organizational attachments, Formalistic task completion) | 2601.52 | 404 | 6.439 | 0.108 | 0.578 | 0.546 | 0.113 |
| **Single factor model** | 3080.328 | 405 | 7.606 | 0.119 | 0.487 | 0.449 | 0.119 |

**Table 3. Reliability and validity.**

| Variables | Factor loading | KMO | Cronbach's α | AVE | CR |
|---|---|---|---|---|---|
| Knowledge | 0.755, 0.795, 0.705, 0.646 | 0.788 | 0.816 | 0.529 | 0.817 |
| Extroversion | 0.713, 0.682, 0.770 | 0.694 | 0.764 | 0.522 | 0.766 |
| Openness | 0.701, 0.719, 0.641 | 0.683 | 0.724 | 0.473 | 0.729 |
| Conscientiousness | 0.748, 0.698, 0.699 | 0.694 | 0.759 | 0.486 | 0.739 |
| Agreeableness | 0.739, 0.718, 0.613 | 0.678 | 0.728 | 0.479 | 0.732 |
| Emotional stability | 0.718, 0.722, 0.607 | 0.682 | 0.726 | 0.468 | 0.724 |
| Avoidant attachment | 0.674, 0.650, 0.643 | 0.670 | 0.693 | 0.430 | 0.694 |
| Anxious attachment | 0.645, 0.705, 0.834, 0.817 | 0.772 | 0.837 | 0.569 | 0.839 |
| Formalistic task completion | 0.778, 0.774, 0.804, 0.732 | 0.819 | 0.854 | 0.597 | 0.855 |

nine-factor model was tested, revealing a good fit with the data. The Chi-square difference test established that the nine-factor model outperformed competing models with fewer factors ($p<0.001$). The study therefore decided to explore the influencing mechanism of the five factors of the Big Five personality, rather than treating individual personality as a singular variable to test the model. Furthermore, based on the confirmatory factor analysis, the Kaiser-Meyer-Olkin (KMO = 0.887) and the significance of the Bartlett test of sphericity (approximate chi-square = 5504.823, and $p$ = 0.000) indicated the suitability of the factor analysis.

To assess the internal consistency and convergent validity of the adopted measures, this study computed composite reliability (CR), AVEs, and factor loading, as shown in Table 3. The majority of CRs exceeded 0.70, meeting the requirement of Hair et al. [67]. The AVEs ranges between 0.430 and 0.597, and there are some AVEs below the recommended level of 0.5, but all CR are higher than 0.6, the convergent validity of the construct is still adequate [68, 73–76]. The standardized factor loading of the items ranged from 0.607 to 0.834, all surpassing the threshold value of 0.500. Additionally, the KMOs were higher than 0.600, and the $p$ values of the Bartlett test of sphericity were less than 0.001. In summary, the construct validity of this study was confirmed. Table 3 also showed that the most of Cronbach's α values were above 0.700, in line with the recommendation by Hair et al. [67]. Furthermore, following the guidance of Hatcher and Stepanski [77], the alpha value of 0.693 for avoidant attachment was deemed acceptable.

To mitigate common method bias, survey data were collected at two time points with a one-and-a-half-month interval. Additionally, the order of items in the questionnaire was deliberately randomized to minimize the effect of respondents' self-attribution. The study also ensured the anonymity and confidentiality of participants. Finally, Harman's single factor was used to measure common method bias. The results indicated that the first factor extracted 24.271% (less than 40%) of the variance, and when all factors were combined, they accounted for 67.830% of the variance, which suggested that the issue of common method bias was effectively controlled in this research.

## Ethics statement

The study was conducted in accordance with the Declaration of Helsinki, and received approval from the Ethics Committee of School of Public Policy and Administration, Nanchang University, China. All participants were provided informed consent with written type. This written informed consent was placed at the beginning of each questionnaire.

## Techniques

The researchers constructed a structural equation model for data analysis, testing the hypotheses developed in the study. This method was chosen due to its suitability for globally measuring the relationships between multiple latent variables. Additionally, a two-way analysis of variance was used to graphically depict the interaction between individual quality and insecure organizational attachment. The tool for data analysis was MPlus 8.3.

## Results

The study aimed to measure both direct and moderating effects in a structural equation model using MPlus 8.3. The theoretical model produced a satisfactory fit for the data ($\chi2$ /df = 1.507, RMSEA = 0.033, CFI = 0.964, TLI = 0.957, SRMR = 0.040). However, the estimation results revealed that the components of individual quality exerted significant effects on formalistic task completion, with only a few moderating effects proving significant. The insignificant paths decrease the fitness of this model. Hence, the theoretical model was bifurcated into two sub-models: the first one for testing direct effects (H1) and the other for testing moderating effects (H2). The model-fitting results are presented in Figs 2 and 3.

As depicted in Fig 2, individual knowledge makes significantly positive effect on formalistic task completion, thus supporting hypothesis H1a. Additionally, all Big Five personalities exhibit significant and positive associations with formalistic task completion, supporting hypothesis H1b. Consequently, H1 was supported totally. A comparative analysis of the effect coefficients reveals that the effects of individual personalities on formalistic task completion surpass those of individual knowledge. Among the personalities, conscientiousness and agreeableness have the strongest effects, followed by extroversion and openness, with emotional stability demonstrating the weakest effect.

Following numerous endeavors, it was discerned that both anxious and avoidant attachment styles exerted a pronounced moderating influence on the interplay between individual knowledge and the completion of formalistic tasks. Specifically, anxious attachment significantly moderated the impact of extraversion on task completion, while avoidant attachment demonstrated a marked moderating effect on the influence of conscientiousness on task completion. That is, insecure organizational attachment could not significantly moderate the

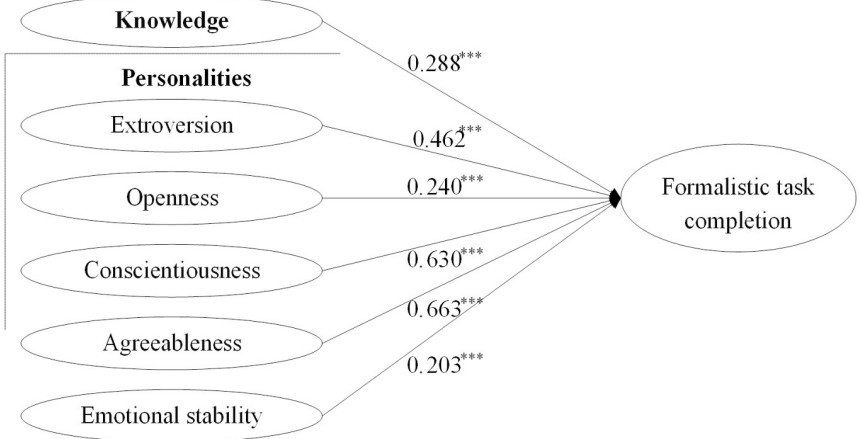

**Fig 2. The estimated direct effects.** ***$p < 0.001$.

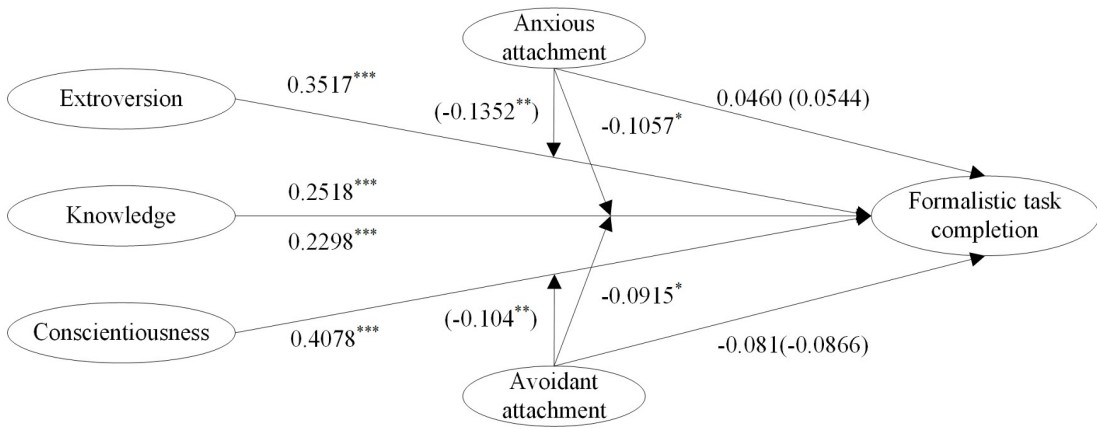

**Fig 3. The estimated moderating effects.** ***$p<0.001$, **$p<0.010$, *$p<0.050$.

positive effects of personalities such as openness, agreeableness, and emotional stability on task completion. Additionally, the interactive effects between avoidant attachment and extroversion, as well as anxious attachment and individual knowledge, were found to be insignificant. Therefore, the insignificant pathways were excluded, and the significant effects were shown in Fig 3. Two results can be gleaned from the Fig 3. The Fig 3 elucidates two key findings. Firstly, it is evident that individual knowledge makes a substantial and positive contribution to the completion of formalistic tasks. Concurrently, both anxious and avoidant attachment styles exert an insignificant and adverse influence on task completion. Furthermore, these attachment styles notably mitigate the positive impact of individual knowledge, thereby providing partial support for Hypothesis H2a. Secondly, the extroverted personality trait of an individual was found to have a substantial and positive influence on the completion of formalistic tasks. In contrast, anxious attachment did not significantly affect the execution of these tasks and, more notably, it was observed to significantly attenuate the positive impact of extroversion. Analogously, a conscientious personality was identified as a robust and positive predictor of formalistic task completion. However, avoidant attachment showed no significant independent effect on task completion. Intriguingly, avoidant attachment was also found to markedly reduce the positive influence of conscientiousness on task execution. Partially supporting hypothesis H2b. In conclusion, hypothesis H2 was partially supported.

The results of the two-way analysis of variance are presented in Fig 4. As shown in the Fig 4, the significant interaction effects between extroversion and anxious attachment, conscientiousness and avoidant attachment, knowledge and avoidant attachment, as well as knowledge and anxious attachment, were once again validated, supporting the hypothesis H2. In Fig 4a–4d, the slopes of the dotted lines, representing low-level moderators, are observed to exceed those of the solid lines, which indicate high-level moderators.. This observation implies that the moderating variables diminish the ability of explanatory variables to predict the explained variables.

## Discussion

Formalistic tasks, serving as a more effective and cost-efficient management tool, are prevalent in modern enterprises and play an important role in maintaining order and advancing the realization of organizational goals in production activities. However, the absence of feedback to employees, involuntary participation, and a lack of recognition for the value of formalistic

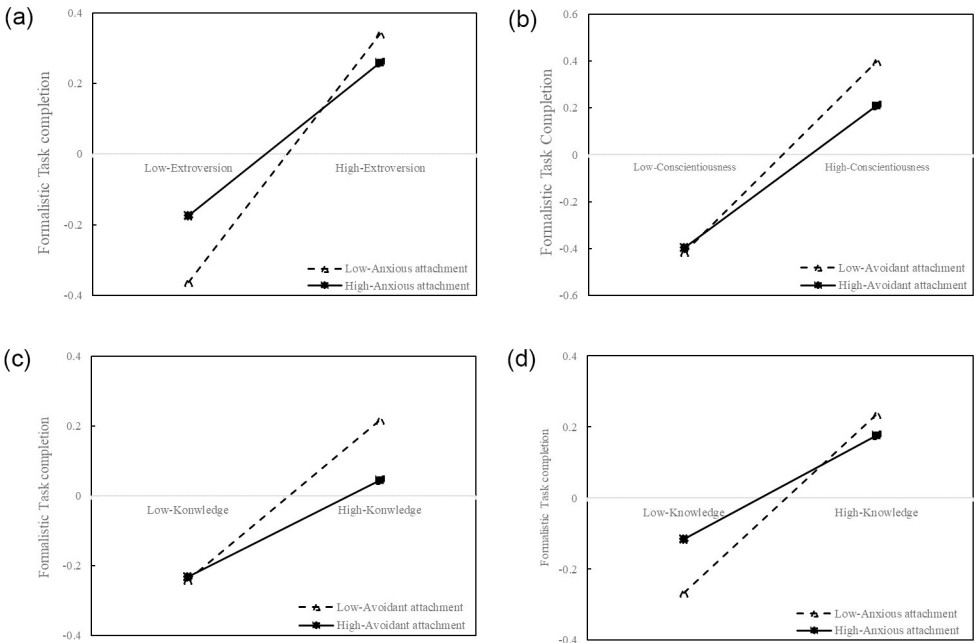

**Fig 4. The interactive effects. Notes**: Fig 4 encompasses four subfigures (Fig 4a-4d) to demonstrate the interactive effects. Fig 4a: Interaction between extroversion and anxious attachment; Fig 4b: Interaction between conscientiousness and avoidant attachment; Fig 4c: Interaction between knowledge and avoidant attachment; Fig 4d: Interaction between knowledge and anxious attachment.

tasks contribute to negative employee perceptions and may trigger negative coping behaviors, such as job burnout. Therefore, it is crucial, based on recognizing the positive value of formalistic tasks, to comprehend how individuals cope with formalistic tasks and appreciate their value. The purpose of the present research is to address this issue by exploring how individuals' quality make effects on formalistic tasks and investigating the role of the organizational environment, specifically organizational attachment, in the mechanisms employed to complete these tasks.

## Theoretical implications

The study extends the application of social cognitive theory to explain the completion mechanism of formalistic tasks. Existing theories on social cognition emphasize that individuals form beliefs about their abilities based on their knowledge, enabling them to effectively utilize this knowledge to improve work performance. In the absence of continuous positive reinforcement, individuals may resist completing tasks. However, formalistic tasks are predominantly perceived by individuals as meaningless, worthless, and subject to bureaucratic negative reinforcement. Therefore, completing formalistic tasks will require not only material and emotional inputs from organizations to employees but will also be influenced by employees' cognitive understanding of the tasks. In this context, whether social cognitive theory can explain the response mechanism of formalistic tasks becomes a challenge. This study confirms that individuals affected by a collectivist culture can participate in tasks even if they do not endorse them, with their completion determined by the individuals' quality, a collection of knowledge, and personalities. Furthermore, while rational thinking affects individuals influenced by environmental factors such as organizational systems and culture, it also underscores

the importance and necessity of an emotional connection with the organization. All in all, social cognitive theory remains applicable to predicting individual responses to formalistic tasks.

The study enhances the comprehension of individual quality theory, attachment theory, and the cognitive-affective system. It introduces the construct of individual quality based on the three-layered onion model, a form of cognitive theory. According to this model, the study posits that individual quality comprises knowledge and personality, utilizing these elements to effectively predict the completion of formalistic tasks. Furthermore, the cognitive-affective system argues that individuals' emotions in an organizational context influence their cognition and behaviors related to tasks and organizations. Attachment theory emphasizes the importance of individuals seeking emotional feedback from the organization or superiors to motivate their engagement in various tasks. This study not only confirms the explanatory power of these theories but also establishes a link between them.

The findings suggest that individuals' behaviors in completing formalistic tasks in organizations are weakened by insecure organizational attachments, indicating that these behaviors are susceptible to the influence of individuals' emotional connections to the organizations. This phenomenon is prevalent in collective cultures. Collectivism, a part of Oriental culture, significantly shapes individual norms and organizational systems. In a collectivist culture, individuals tend to identify themselves as organizational members, paying attention to their relationships with others in the same organization. When individual interests clash with organizational interests, individuals willingly make concessions for the collective good [78]. Organizational attachment theory emphasizes that individuals' different attachment styles affect how they approach work and their willingness and ability to exert effort [79, 80]. The formalistic task concerned in this study is one of the derivatives of this cultural phenomenon. That is, individuals need to establish a positive relationship with the organization so that, even when faced with tasks they may dislike, they can still be done through an intimate emotional and psychological connection to the organization. This perspective aligns with Akroyd and Kober's work [81], where they proposed that people are reluctant to implement informal controls unrelated to organizational culture because it would have a negative effect. However, organizations can overcome this reluctance by fostering a workplace where employees develop strong emotional attachments to each other.

## Practical implications

Firstly, the study of social cognitive theory focuses on the dynamic interaction between individuals' internal factors, environmental factors, and behaviors. It holds that individuals are not just reactive organisms shaped by the external environment but are also self-organized and self-adjusted learners. It is crucial for understanding the dynamic competitiveness and work performance in modern organizations, offering a unique psychological and developmental perspective on work-related abilities. For example, Wu and Parker [21] noted that secure-base support from leaders can predict positive work behaviors in employees. This prediction occurs by increasing their role span self-efficacy and autonomous motivation, explaining these behaviors by satisfying individual needs associated with insecure attachment. Therefore, for managers, comprehending the secure attachment between employees and organizations is vital for understanding how to motivate proactive behaviors in individuals.

Secondly, the positive effect of individual personalities on the completion of formalistic tasks indicates the importance of individuals acquiring more knowledge and cultivating positive personalities. This is crucial for enhancing their sense of self-efficacy, thus instilling greater confidence in the completion of work tasks. This observation implies that managers should

pay attention to individual knowledge management, actively contributing to its improvement when evaluating employees' work performance.

Finally, organizational attachment has practical significance for individuals' traits and work performance. The negative effect of insecure organizational attachment in moderating the relationship between individual quality and the completion of formalistic tasks suggests that avoidant attachment hampers individuals from using their knowledge and the strength of conscientiousness to improve work performance, while anxious attachment impedes performance, for those with extroverted personality and well-educated individuals. To foster employees' completion of formalistic tasks, individuals with limited knowledge can cultivate a sense of security and belonging in the organization by strengthening their connection to it. This approach can create a more relaxed and secure working environment, consequently improving their cognitive abilities and enhancing work performance. Moreover, introverted individuals may benefit from a harmonious organizational atmosphere, relying on it to foster a sense of relaxation and tranquility, ultimately improving their task performance at work.

## Limitations and future directions

The research is not without limitations. First, despite the attempt to collect survey data at two different time points, the difficulty lies in supporting statistical inferences regarding causal relationships. Future research suggested to use a longitudinal study design to replicate the findings of the current study. Second, the scales measuring knowledge and formalistic task completion warrant further testing in subsequent studies. Third, the study's limitation to a sample of university students diminishes the generalizability of the findings. Future research should aim to include more diverse samples to bolster the robustness of these findings. Finally, the study solely engages in an exploration of the relationship between individual quality and formalistic task completion. In subsequent research, the authors plan to deepen their research by introducing intermediate conditions (such as working competence and organizational satisfaction) to establish a more comprehensive framework. This involves refining measurement tools and expanding the sample size.

## Conclusion remark

Formalistic tasks represent rational and collective actions crucial for hierarchical bureaucratic organizations, facilitating the realization of coordinated efforts, fostering organizational order, and achieving defined organizational objectives. This study aimed to explore the influential factors shaping the completion of formalistic tasks and the effect of the psychological bond between individuals and organizations on work performance. Results highlighted the significance of both individual knowledge and personalities in significantly influencing task completion, with insecure organizational attachments moderating this relationship. This study contributed to enriching social cognitive theory. In the future, the authors intend to conduct further extensive research on formalistic task management, delving deeper into revealing the mechanisms underlying their generation and completion.

## Supporting information

**S1 File. The questionnaire of the study.**
(DOCX)

**S2 File. Data.**
(XLSX)

## Acknowledgments

The authors gratefully acknowledge the helpful comments and suggestions of the editor and the reviewers, which have improved the presentation.

## Author Contributions

**Conceptualization:** Dengke Yu.

**Data curation:** Wenjun Wu.

**Formal analysis:** Wenjun Wu, Huan Xiao, Dengke Yu.

**Funding acquisition:** Dengke Yu.

**Investigation:** Wenjun Wu.

**Methodology:** Wenjun Wu, Dengke Yu.

**Project administration:** Dengke Yu.

**Resources:** Huan Xiao.

**Software:** Wenjun Wu.

**Supervision:** Dengke Yu.

**Validation:** Wenjun Wu, Huan Xiao, Dengke Yu.

**Visualization:** Huan Xiao.

**Writing – original draft:** Wenjun Wu.

**Writing – review & editing:** Huan Xiao, Dengke Yu.

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
