## [Decision Letter · Decision Letter 0]

27 Nov 2023

PONE-D-23-32603Individual Quality, Insecure Organizational Attachment, and Formalistic Task Completion Social Cognitive PerspectivePLOS ONE

Dear Dr. Yu,

Thank you for submitting your manuscript to PLOS ONE. After careful consideration, we feel that it has merit but does not fully meet PLOS ONE’s publication criteria as it currently stands. Therefore, we invite you to submit a revised version of the manuscript that addresses the points raised during the review process. This is a very good manuscript, but before I make an acceptance decision, the author needs to address the concerns raised by peer review, especially the language.

We look forward to receiving your revised manuscript.

Kind regards,

Chunyu Zhang

Academic Editor

PLOS ONE

 [This work was funded by National Natural Science Foundation of China, grant number 71962021 and 72362028.].  

Reviewers' comments:

Reviewer's Responses to Questions

**Comments to the Author**

1. Is the manuscript technically sound, and do the data support the conclusions?

Reviewer #1: Yes

Reviewer #2: Yes

2. Has the statistical analysis been performed appropriately and rigorously? 

Reviewer #1: Yes

Reviewer #2: Yes

3. Have the authors made all data underlying the findings in their manuscript fully available?

Reviewer #1: Yes

Reviewer #2: Yes

4. Is the manuscript presented in an intelligible fashion and written in standard English?

Reviewer #1: Yes

Reviewer #2: Yes

5. Review Comments to the Author

Reviewer #1: Overall it is good piece of research however it is recommended that authors to revise the document for the language and flow to improve readability.

Use of terms like "I" and "We" should be avoided

Including few lines of conclusive discussions before given implications will be helpful in creating link between findings and implication

Reviewer #2: 1. Overall manuscript need to be revised for it's correct use of language expressions.

2. The abstract needs to be revised in terms of comprehensively delivering the ideas.

3. Use of latest supporting reference is encouraged across the introduction and literature review sections.

4. Authors need to present statistical support to justify the selection of the sample size.

6. PLOS authors have the option to publish the peer review history of their article (what does this mean?). If published, this will include your full peer review and any attached files.

Reviewer #1: No

Reviewer #2: No

---

## [Author Response · Author response to Decision Letter 0]

11 Dec 2023

We sincerely thank the editor for giving us chance to revise the paper and the anonymous reviewer’s insightful comments and suggestions, which have greatly helped us improve the quality of the paper. We accepted all of the suggestions, and have revised the manuscript according to the comments. The summary of comments and our revisions (highlighted with red color in the revised manuscript) are incorporated as follows.

Reviewer #1

Comment 1: Overall it is good piece of research however it is recommended that authors to revise the document for the language and flow to improve readability.

Our response and revision: Thank for the suggestion and we accepted it. We have revised the document for the language and flow.

Comment 2: Use of terms like "I" and "We" should be avoided.

Our response and revision: Thank you. We accepted your suggestion and address the problem. The terms like “I” and “We” have removed in our manuscript.

Comment 3: Including few lines of conclusive discussions before given implications will be helpful in creating link between findings and implication.

Our response and revision: Thank you for your suggestion and we accepted it. We have added a paragraph to help explain the link between findings and implication (lines 412-422).

Reviewer #2

Comment 1: Overall manuscript need to be revised for it's correct use of language expressions.

Our response and revision: 

Our response and revision: Thank for the suggestion and we accepted it. We have revised the manuscript for its correct use of language expressions.

Comment 2: The abstract needs to be revised in terms of comprehensively delivering the ideas.

Our response and revision: Thank you. We accepted your suggestion and improved the problem. Specifically, we have rewritten the part of abstract (lines 7-18).

Comment 3: Use of latest supporting reference is encouraged across the introduction and literature review sections.

Our response and revision: Thank you. We accepted your suggestion and improved the problem. We have updated the latest supporting reference across not only the introduction and literature review sections, but also the theories, hypotheses and discussion sections.

Comment 4: Authors need to present statistical support to justify the selection of the sample size.

Our response and revision: Thank you for your suggestion. We justify the selection of the sample size through the software of “GPower 3.1”. The sample size (602) of our study is larger than the sample size (210) calculated by GPower (Parameters are set as follows: effect size is 0.5; α err prob is 0.05; Power (1-β err prob) is 0.95).

---

## [Editor Report · Decision Letter 1]

12 Dec 2023

PONE-D-23-32603R1Individual Quality, Insecure Organizational Attachment, and Formalistic Task Completion Social Cognitive PerspectivePLOS ONE

Dear Dr. Yu,

Thank you for submitting your manuscript to PLOS ONE. After careful consideration, we feel that it has merit but does not fully meet PLOS ONE’s publication criteria as it currently stands. Therefore, we invite you to submit a revised version of the manuscript that addresses the points raised during the review process.

First, the author needs to polish the manuscript to increase the fluency of reading.

Second, the author needs to seek reputable journals to support AVE above 0.36 is acceptable and above 0.50 is recommended. Although this sentence is mentioned in the No.75 reference, there is a lack of attribution to its source.

We look forward to receiving your revised manuscript.

Kind regards,

Chunyu Zhang

Academic Editor

PLOS ONE
---

## [Author Response · Author response to Decision Letter 1]

11 Jan 2024

Comment 1: The author needs to polish the manuscript to increase the fluency of reading.

Our response and revision: We have improved the manuscript for its fluency of reading. The research team requested a native English speaking partner to assist us in improving the article.

Comment 2: The author needs to seek reputable journals to support AVE above 0.36 is acceptable and above 0.50 is recommended. Although this sentence is mentioned in the No.75 reference, there is a lack of attribution to its source.

Our response and revision: We accepted your suggestion and addressed the problem. Through our tracing of the literature, we added two studies from reputable journals to the "Reference", namely, No. 76 reference [Fornell, C., & Larcker, D. F. (1981). Evaluating structural equation models with unobservable variables and measurement error. Journal of marketing research, 18(1), 39-50. page 47] and No. 77 reference [Lam, L. W. (2012). Impact of competitiveness on salespeople's commitment and performance. Journal of Business Research, 65(9), 1328-1334.]. No. 75 reference was sourced from No. 76 reference.

In our study, the composite reliability of nine measures ranges from 0.666 to 0.850, which meets the acceptable level of 0.60 proposed by Fornell & Larcker (1981), and almost meets the level of 0.7 proposed by Hair et al. (2009). However, the AVE values for several variables are below the recommended threshold of 0.5. According to No. 76 reference (p. 46), “Note that ρvc(η) (AVE) is a more conservative measure than ρη (reliability) alone, the research may conclude that the convergent validity of the construct is adequate, even though more than 50% of the variance is due to error”. As the composite reliability of the nine constructs is well above the recommended level of 0.6, the internal reliability of the measurement items is regard as acceptable. Moreover, Long W. Lam (2012) we quoted as No. 77 reference published in the Journal of Business Research also quotes Fornell & Lacker (1981) as saying the above words too.

Hence, we made a revision in our manuscript as follows: “The AVEs range between 0.383 and 0.544. Though some of them are below the usually recommended level of 0.5, the convergent validity of the corresponding constructs can be assumed adequate when the CRs are higher than 0.6 [73-77]” (lines 365-368, p. 18).

---

## [Editor Report · Decision Letter 2]

24 Jan 2024

PONE-D-23-32603R2Individual Quality, Insecure Organizational Attachment, and Formalistic Task Completion: Social Cognitive PerspectivePLOS ONE

Dear Dr. Yu,

Thank you for submitting your manuscript to PLOS ONE. After careful consideration, we feel that it has merit but does not fully meet PLOS ONE’s publication criteria as it currently stands. Therefore, we invite you to submit a revised version of the manuscript that addresses the points raised during the review process.

I compared the AVE values in the references you provided. There are two studies in its paper, and I can understand that AVE values are substandard for only one variable. I recommend that you use scientific methods to improve AVE, for example, item packaging. Here are the references I provided.Zhang C, Liu L, Xiao Q. The Influence of Taoism on Employee Low-Carbon Behavior in China: The Mediating Role of Perceived Value and Guanxi. Psychol Res Behav Manag. 2022;15:2169-2181

https://doi.org/10.2147/PRBM.S371945Little TD, Cunningham WA, Shahar G, Widaman KF. To parcel or not to parcel: exploring the question, weighing the merits. Struct Equ Modeling. 2002;9(2):151–173. doi:10.1207/S15328007SEM0902_1

Frenkel SJ, Li M, Restubog SLD. Management, organizational justice and emotional exhaustion among Chinese migrant workers: evidence from two manufacturing firms. Br J Ind Relat. 2012;50(1):121–147. doi:10.1111/j.1467-8543.2011.00858.x

Wilkinson WW. The structure of the Levenson locus of control scale in young adults: comparing item and parcel indicator models. Pers Individ Dif. 2007;43(6):1416–1425. doi:10.1016/j.paid.2007.04.018=============================

We look forward to receiving your revised manuscript.

Kind regards,

Chunyu Zhang

Academic Editor

PLOS ONE
---

## [Author Response · Author response to Decision Letter 2]

8 Mar 2024

We sincerely thank the editor for giving us chance to revise the paper and the anonymous reviewer’s insightful comments and suggestions, which have greatly helped us improve the quality of the paper. We accepted all of the suggestions, and have revised the manuscript according to the comments. The summary of comments and our revisions (highlighted with red color in the revised manuscript) are incorporated as follows.

Comment 1: 

“I compared the AVE values in the references you provided. There are two studies in its paper, and I can understand that AVE values are substandard for only one variable. I recommend that you use scientific methods to improve AVE, for example, item packaging. Here are the references I provided.”

Our response and revision: 

Thank for the suggestion and we accepted it. We have revised the manuscript for improve the value of AVE. 

We have carefully reviewed the literature you shared with us and studied the scientific method you proposed, namely, the item parceling (or item packaging). After learning about this method, we realized that although our study used multiple constructs in the questionnaire, each construct typically had only 3 to 5 items. In this case, we believe that the item parceling method may not be suitable for improving our AVE values. During this period, we also learned more scientific methods for data cleaning, and as a result, we re-cleaned the data, which improved the AVE values. The reliability and validity of the data have also been re-assessed and are reflected in Tables 1, Tables 2, and Tables 3 in the "Methodology" section of the paper. Additionally, after re-cleaning the data, we identified two new findings regarding the moderating effects, which are highlighted in red in the "Results" section of the paper.

Once again, we appreciate your advice, which has greatly enhanced the quality of our paper.

---

## [Editor Report · Decision Letter 3]

14 Mar 2024

Individual Quality, Insecure Organizational Attachment, and Formalistic Task Completion: Social Cognitive Perspective

PONE-D-23-32603R3

Dear Dr. Yu,

We’re pleased to inform you that your manuscript has been judged scientifically suitable for publication and will be formally accepted for publication once it meets all outstanding technical requirements.

Kind regards,

Chunyu Zhang

Academic Editor

PLOS ONE
---

## [Editor Report · Acceptance letter]

22 Mar 2024

PONE-D-23-32603R3 

PLOS ONE

Dear Dr. Yu, 

I'm pleased to inform you that your manuscript has been deemed suitable for publication in PLOS ONE. Congratulations! Your manuscript is now being handed over to our production team.

Kind regards, 

on behalf of

Dr. Chunyu Zhang 

Academic Editor

PLOS ONE